# Physicochemical and Rheological Characteristics of Commercial and Monovarietal Wheat Flours from Peru

**DOI:** 10.3390/foods12091789

**Published:** 2023-04-26

**Authors:** Ivan Best, Alan Portugal, Sandra Casimiro-Gonzales, Luis Aguilar, Fernando Ramos-Escudero, Zoila Honorio, Naysha Rojas-Villa, Carlos Benavente, Ana María Muñoz

**Affiliations:** 1Instituto de Ciencias de Los Alimentos y Nutrición, Universidad San Ignacio de Loyola (ICAN-USIL), Campus Pachacamac, Lima 15823, Peru; aportugal@usil.edu.pe (A.P.); scasimiro@usil.edu.pe (S.C.-G.); laguilar@usil.edu.pe (L.A.); dramos@usil.edu.pe (F.R.-E.); amunoz@usil.edu.pe (A.M.M.); 2Carrera de Medicina Humana, Facultad de Ciencias de la Salud, Universidad San Ignacio de Loyola, Lima 15024, Peru; 3Facultad de Bromatología y Nutrición, Universidad Nacional José Faustino Sánchez Carrión, Lima 15136, Peru; zhonorio@unjfsc.edu.pe; 4Centro Internacional de Investigación para la Sustentabilidad, Universidad Nacional de Cañete, Lima 150501, Peru; n_rojas@undc.edu.pe; 5Facultad de Farmacia y Bioquímica, Universidad Nacional San Luis Gonzaga de Ica, Ica 11004, Peru; carlos.benavente@unica.edu.pe

**Keywords:** wheat, flour, quality, gluten, gliadin, glutenin, physicochemical, rheology, regression model

## Abstract

In Peru, wheat (*Triticum aestivum* L.) is one of the main resources in the food industry; however, due to its low harvested area, it is the second most imported cereal. The quality of wheat flour was studied to verify that it has desirable characteristics for the preparation of bakery products. The quality of commercial and monovarietal wheat flours was assessed by measuring their physicochemical and rheological parameters, as well as the gluten content and wheat protein fractions. Eight commercial wheat flours and four monovarietal wheat flours (Barba negra, Candeal, Espelta, and Duro) from Peru were evaluated. Commercial wheat flours presented significantly higher levels of protein and gluten index compared to monovarietal wheat flours (*p* < 0.05). Between both groups, no significant differences were observed in the content of wet and dry gluten. Interestingly, monovarietal wheat flours presented a higher percentage of gliadins and albumins/globulins, as well as lower levels of glutenin, compared to commercial wheat flours (*p* < 0.05). According to the logistic regression models, the baking strength (*W*) was the most important parameter to evaluate the quality of commercial and monovarietal wheat flours. Our results show that monovarietal wheat flours show a lower quality compared to commercial wheat flours.

## 1. Introduction

In South America, apart from corn, wheat is one of the crops that is produced on a larger scale. During the period of 2020–2025, it is estimated that the wheat market will reach a compound annual rate (CAGR) of 4.5% [1]. In 2031, Argentina is forecast to be the largest wheat producer in Latin America, with 23.44 million metric tons, followed by Brazil, with almost 7.5 million metric tons, while Peru is projected to produce just 0.24 million metric tons of wheat [2].

In Peru, wheat (*Triticum aestivum* L.) is part of the basic consumption of the population, but its harvested area is carried out on a reduced scale. In 2019 and 2020, the area harvested for wheat in Peru only reached 120,634 and 114,344 hectares, respectively [3].

Wheat flour is the main resource for the production of bread, pasta, and cookies. In Peru, the cultivation of wheat used for the elaboration of farinaceous products cannot cover the internal demand that annually amounts to 2 million metric tons. Only a small part of the wheat produced annually in Peru (218 thousand metric tons) is destined for the milling industry. Likewise, it has been reported that wheat flour from Peru has a low bakery quality [4], which is why the demand for this product is supplied mainly by imports from Canada, the United States, Russia, and Argentina (around 90% of the market total) [5]. In 2020, imports of durum wheat came entirely from Canada (100%), while imports of other wheats came from Canada (77.2%), the United States (9.84%), and Argentina (12.96%) [6].

The quality of wheat flour can be determined by different physicochemical tests, such as the quantification of protein, gluten, moisture, and ash, the measurement of α-amylase activity by falling numbers, as well as the evaluation of rheological properties to obtain different alveographic, farinographic, mixographic, and extensographic measurements. In general, the quality of wheat flour for bread making is evaluated by the amount of protein and the quality of gluten [7]. Wheat flour containing high protein and high-quality gluten is used for making bread, while flour with lower protein is mainly used for pastries or cakes [8,9].

Based on their solubility, wheat proteins are classified into four so-called Osborne fractions: albumins, globulins, gliadins, and glutenins [10]. The albumin/globulin fraction represents 20–25% of wheat proteins, while gliadins and glutenins account for 75–90% of wheat proteins (Figure 1). Gluten is made up of 90% protein, 8% fat, and 2% carbohydrate, and is a complex mixture of hundreds of related but distinct proteins, mainly made up of gliadins and glutenins [11].

Among the most important parameters to assess the quality of wheat flour, the protein concentration in the grain as well as its composition, determined by the levels of albumins, globulins, gliadins, and glutenins, are the most relevant to produce wheat flour with high bakery quality. Likewise, environmental factors such as the management of soil fertilization with nitrogen can influence the content and composition of wheat proteins [12]. In general, the higher the protein content of a flour, the greater the gluten formation. Different varieties of wheat vary in their protein content and in the composition and distribution of gluten proteins [13].

Gluten plays a key role in determining the baking quality of wheat by providing water absorption capacity, cohesiveness, viscosity, and elasticity to dough [14]. The gluten formed by the proteins gliadin and glutenin, when combined with water, forms a network capable of retaining the carbon dioxide released during fermentation. This quality is determined by rheological tests that allow prediction of the behavior of the flour during the baking process and the characteristics that the final products will have [15]. Likewise, the gluten index (*GI*) is a criterion that defines whether the quality of the gluten is weak (*GI* < 30%), normal (*GI* = 30–80%), or strong (*GI* > 80%). A study shows that wheats with similar protein content can be classified according to gluten index values [16].

The objective of this study was to evaluate the quality of commercial and monovarietal wheat flours used mainly in the Peruvian baking industry by assessing their physicochemical and rheological parameters, as well as gluten content and wheat protein fractions (albumins/globulins, gliadins, and glutenins), and thus estimate their usefulness in the preparation of bakery products.

## 2. Materials and Methods

### 2.1. Chemicals

The chemicals were purchased from Sigma-Aldrich (G5003, St. Louis, MO, USA). They were: CaO, HCl, HgO, K_2_SO_4_, Na_2_SO_4_, NaCl, H_2_SO_4_, NaOH, ethanol, diethyl ether petroleum ether, trifluoroacetic acid, and acetonitrile. The solvents were ACS reagent grade or higher.

### 2.2. Collection of Commercial and Monovarietal Wheat Flours

During the study, eight commercial wheat flours purchased from different milling companies in Peru and four flours obtained from monovarietal wheat (Barba negra, Espelta, Candeal, and Duro) were selected, which have the greatest economic importance in Peru (Table 1). The monovarietal wheats were obtained from each of the varieties mentioned, without any type of grain mixture, and all came from the province of Acobamba, Department of Huancavelica, Peru.

For the preparation of flours from monovarietal varieties of wheat, the grains were ground in a CD1 mill (Chopin Technologies, Paris, France) to a grain size no greater than 160 µm. The obtained flours were kept at −20 °C in vacuum-sealed bags (non-permeable to oxygen and moisture) until the analyzes were performed [17].

### 2.3. Measurement of Wheat Properties

#### 2.3.1. Proximate Analysis

Moisture, ash, fat, and protein content were analyzed according to AOAC methods 925.10, 923.03, 922.06, and 920.87, respectively [18]. The crude fiber was determined by the AOCS method Ba 6-84 [18]. Carbohydrate content was determined as a difference of mean values: 100 − sum of moisture, ash, fat, and protein content.

#### 2.3.2. Flour Color Measurements

Flour color measurements were carried out by image analysis [19]. Approximately 10 g of flour sample was placed in a Petri dish. Image acquisition was developed in a LED portable photo studio (40 cm^3^) (Puluz Technology Limited, Shenzhen, China) containing 2 light boards each with 32 PCs LED lights (output power: 30 W, color temperature: 5500 K). The photographs were acquired using a digital camera (Canon, Power Shot SX60 HS, full HD 65X optical zoom, Tokyo, Japan). The photographic images were stored on a memory card (SanDisk Ultra, 128 GB) and subsequently analyzed on a laptop using ImageJ−1.51k plugins (National Institutes of Health by Wayne Rasband, Bethesda, MD, USA). The RGB values obtained were converted to system *L**, *a**, *b** using the online software ColorHexa.com interface. Color parameters for C**ab* (chroma) and h*ab* (hue angle) were calculated [20] using the following equations:(1)$$C*ab=\mathrm{sqrt}\;\left(a{*}^{2}+b{*}^{2}\right)$$
(2)$$hab=arctan\left(\frac{a*}{b*}\right)$$

#### 2.3.3. Gluten Analysis

Wet gluten, dry gluten, and gluten index were analyzed according to the AACC method 38-12.02 [21] using the Glutomatic 2200 (Perten 153 Instruments, Stockholm, Sweden).

#### 2.3.4. Falling Number

The α-amylase activity was determined according to the AACC method 56-81B [21] using the Hagberg descending numbers kit (Partens model no. 1500, Instrumentvägen 31 SE-126 53, Stockholm, Sweden).

#### 2.3.5. Rheological Analysis

The alveograph parameters were determined by an alveo-consistograph (Chopin, Paris, France) using the AACC 54-30.02 method [21]. Each Alveograph curve was analyzed for the following parameters: *P*, the maximum pressure needed to blow the dough bubble, expresses dough resistance; *L*, length of the curve, expresses dough extensibility; *P/L*, the configuration ratio of the Alveograph curve; and *W*, the baking strength (surface area of the curve) [22]. Likewise, the water absorption capacity (*WAC*) was evaluated using the AACC 54-50.01 method [21].

#### 2.3.6. Proteins RP-HPLC for Gluten

A Chromaster High Performance Liquid Chromatograph (HPLC) (Hitachi, Tokyo, Japan) with EZChrom Elite Ver. 3.2.1 software, an autosampler, and a BIOshell™ A400 Protein column C18 (L × I.D. 15 cm × 3 mm, 3.4 μm particle size, Supelco, Bellefonte, PA, USA) were used. The mobile phase consisted of trifluoroacetic acid (TFA) (0.1%, *v*/*v*) in water (A) and TFA (0.1%, *v*/*v*) in acetonitrile (B).

The extraction of albumins/globulins, gliadins, and glutenins was carried out according to Pronin et al. [11]. For the determination of wheat protein fractions, the gradient profile was 0 min, 0% B; 0.5 min, 20% B; 7 min, 60% B; 7.1–11 min, 90% B; and 11.1–17 min, 0% B for albumins/globulins and 0 min, 0% B; 0.5 min, 24% B; 20 min, 56% B; 20.1–24.1 for gliadins and glutenins. Flow and temperature were 0.2 mL/min and 60 °C, respectively. The injection volume was 10 μL for gliadins and 20 μL for albumins/globulins and glutenins. Detection was performed at 210 nm using a diode array detector (HPLC-DAD 300). Data were processed using OpenLAB CDS software (Agilent Technologies, Santa Clara, CA, USA). Prolamin Working Group (PWG)-gliadin (2.5 mg/mL in 60% ethanol) was used for external calibration and calculation of protein content [23].

#### 2.3.7. Statistical Analysis

The results were analyzed using Statistical Package for the Social Sciences (SPSS) software for Windows version 28.0 (SPSS, Inc., Chicago, IL, USA). Data represents the mean ± standard deviation of duplicate samples. Differences between groups were evaluated using a two-sample *t*-test. The correlation between the different variables was evaluated using the Pearson method. These results were considered significant at *p* < 0.05 and *p* < 0.01. The backward stepwise multiple regression was carried out to evaluate if the physicochemical characteristics were associated with the quality parameters of commercial and monovarietal wheat flours from Peru. Using this method, the best subgroup of independent variables with the largest adjusted R^2^ multiple correlation coefficients that contributed to predicting the dependent variable was identified. The dependent variables were dry gluten, gluten index, and falling number. A level of 0.05 was used to assess the statistical significance of the predictive models.

## 3. Results and Discussion

### 3.1. Physicochemical Characteristics

Table 2 shows the proximal analysis of commercial and monovarietal wheat flours. The samples evaluated in the study presented a moisture, ash, fat, protein, carbohydrate, crude fiber, and total energy content that ranged from 13.00% to 16.21%, 0.41% to 0.71%, 0.24% to 1.55%, 6.26% to 12.03%, 71.64% to 77.92%, 0.09% to 0.20%, and 333.94 to 351.13 Kcal/100 g, respectively.

Monovarietal wheat flours had significantly higher moisture compared to commercial wheat flours (*p* < 0.01). Moisture is one of the parameters that most affects wheat quality, which is associated with relative humidity at harvest and during storage [9]. Moisture content in flours from Sudanese wheat cultivars ranged from 10.21% (Dongla) to 13.13% (Medani) [24]. Similar moisture levels were found in our commercial wheat flours. In general, moisture levels of 14% or less allow wheat flours to be stable at room temperature and limit the growth of microorganisms [25], which was not found in our monovarietal wheat flours.

Monovarietal wheat flours showed a significantly higher ash content compared to commercial wheat flours (*p* < 0.01). Ash content is related to flour contamination with bran particles. During milling, an inadequate separation between the bran and the germ of the endosperm would cause the bran to give a darker color to the products [26]. Likewise, the ash content is related to the amount of minerals present in the grain and directly affects the color of the flour [27]. Previous studies show that the ash content of wheat flours ranged from 0.29% to 0.57% [28] and 0.49% to 0.61% [29]. Another study in Sudan showed that the ash content of twenty wheat cultivars varied between 0.47% to 0.85% (Medani) [24]. Similar results were found in the commercial and monovarietal wheat flours evaluated in the present study. According to Makawi et al. [9], the variation in ash content between different flours is associated with the wheat production site.

Furthermore, monovarietal wheat flours presented higher crude fiber levels compared to commercial wheat flours (*p* < 0.05). No significant differences in carbohydrate levels were found between both varieties of wheat flours. Carbohydrate content, particularly fiber and starch levels, has been reported to influence the water absorption capacity (*W*) of flours. This parameter is associated with bakery quality. Although a high *WAC* could give a smaller volume to the bread, the form ratio would be adequate due to the greater consistency of the dough [30,31].

Commercial wheat flours presented significantly higher protein levels compared to monovarietal wheat flours (*p* < 0.05). The protein content is an important parameter to determine the quality of wheat, which is associated with genetic and non-genetic factors. Flour with low protein content is used for the preparation of cakes and cookies [32,33]. In Indian wheat cultivars, flour protein levels ranged from 8.65% to 12.02% [34] and 8.26% to 12.85% [29]. In the present study, in 75% of the commercial wheat flour the protein content was higher than 9%, while in only 25% of the monovarietal wheat flours the protein content was higher than this level.

### 3.2. Flour Color Characteristics

The chromatic properties of commercial and monovarietal wheat flours are shown in Table 3. *L**, *a**, *b**, *C***ab*, and h*ab* values of commercial and monovarietal wheat flours ranged from 81.17 to 88.89, −3.09 to −1.67, 8.66 to 14.46, 8.94 to 14.61, and 74.31 to 81.66, respectively.

Monovarietal wheat flours presented significantly higher values of *L** compared to commercial wheat flours (*p* < 0.01). *L** indicates the lightness of the flours [26]. In commercial and monovarietal wheat flours, the value of *L** was negatively correlated with protein levels, although statistically significant differences were not found.

No significant differences were observed in the *a** values between both varieties of wheat flour. The *a** indicates the red/green value, which means that flours with higher redness show a higher *a** value [26]. In the different flours evaluated, negative values of *a** were found. In commercial and monovarietal wheat flours, the value of *a** showed a significant negative correlation with the ash content (r = −0.711, *p* < 0.01).

Monovarietal wheat flours showed a higher *b** value compared to commercial wheat flours (*p* < 0.01). Here, *b** denotes the yellow/blue value of flours [26]. A higher value of *b** is associated with the content of xanthophylls in wheat flour [26]. Furthermore, the *C**ab and *h*ab values were significantly higher in commercial wheat flours compared to monovarietal wheat flours (*p* < 0.01). In general, the color differences between different varieties of wheat flour could be due to their xanthophyll and ash content, as well as the particle size distribution [35].

### 3.3. Gluten and Falling Number Characteristics

Table 4 shows the protein quality of commercial and monovarietal wheat flours. The wet and dry gluten content ranged from 12.13 to 37.33% and 4.33 to 12.28%, respectively. In the same trend, regarding the higher levels of protein presented by commercial flours, the latter showed higher levels of wet gluten and dry gluten compared to monovarietal flours; however, statistically significant differences were not observed between both types of wheat flours.

The gluten levels in the grain are related to the total protein content, wet, and dry gluten [33]. Gluten proteins primarily determine the processing quality of wheat. Proteins are the main components of dry gluten; therefore, this parameter reflects the characteristics of the gluten content. The protein content in gluten between different wheat varieties is associated with genetic factors, as well as environmental conditions [36]. Our results are comparable to those obtained on different wheat varieties grown in India. In Australian wheat varieties, wet gluten ranged from 31.65% (Longreach Orion) to 41.96% (EJA 2248), while in Indian varieties, wet gluten ranged from 5.48% (HS490) to 13.32% (C306) [33].

Among the different wheat flours evaluated in the present study, the gluten index varied between 40 and 99%, which was significantly higher in commercial wheat flours compared to monovarietal wheat flours (*p* < 0.01). The gluten index is associated with the degree of elasticity and extensibility of the flour; the higher the gluten index, the stronger the gluten obtained [33]. Similar to other studies, the wheat varieties evaluated in our study ranged from weak to strong gluten. In Indian wheat varieties, the gluten index ranged from 27% (K307, HW2004) to 100% (DPW 621-50, WH 1080, HD 2987, HUW 468, PBW550), while in Australian varieties, the gluten index ranged from 42% (Longreach Orion) to 94% (Yitpi) [33]. According to the gluten index, our commercial wheat flours presented strong gluten, while monovarietal wheat flours showed normal gluten.

Regarding the α-amylase activity of the wheat flours included in the study, the falling number ranged from 135 to 483 s. Falling number levels were significantly lower in monovarietal wheat flours compared to commercial wheat flours (*p* < 0.01). For wheat flour, the optimal value of the falling number is around 250 s. Falling numbers between 105 and 220 s correspond to flours with high α-amylase activity and require correction, while wheat flours with falling numbers greater than 350 s have low α-amylase activity and require supplementation with amylolytic enzyme or malted grain [37]. In commercial and monovarietal wheat flours, falling number was significantly correlated with gluten index (r = 0.780, *p* < 0.01), *W* (r = 0.819, *p* < 0.01) and *b** value (r = 0.819, *p* < 0.01).

When the protein fractions of the different wheat flours were evaluated (Figure 2), the levels of albumins/globulins, gliadins, and glutenins ranged from 18.70 to 27.32%, 27.14 to 43.68%, and 31.94 to 54.03%, respectively. Interestingly, monovarietal wheat flours had a significantly higher percentage of gliadins and albumins/globulins (*p* < 0.05), as well as lower glutenin levels compared to commercial wheat flours (*p* < 0.01).

In wheat, gliadins and glutenins are normally found in a 50/50 ratio. Both proteins play an important role in determining the baking quality of wheat flours. A balance between gliadins and glutenins, as well as total protein content, determines that the flours have a medium strength and are used for baking. The gliadins contribute mainly to the viscosity and extensibility of the dough, while the glutenins are related to the resistance and elasticity of the dough.

Therefore, wheat flours that have a higher proportion of glutenins are stronger and more tenacious, while flours that have a higher proportion of gliadins are more viscous and extensible [38].

### 3.4. Rheological Characteristics

In the bakery industry, to determine the resistance and extensibility of a flour dough, many rheological tests such as the pharyngograph, the extensograph, and the alveograph are used to predict the usability of the flour and the quality of the final products without the need for perform a real baking test [33,38]. Table 5 shows the rheological analysis of commercial and monovarietal wheat flours. *WAC*, *P*, *L*, *P/L*, and *W* levels ranged from 35.20 to 58.70%, 41 to 140 mm, 17 to 148 mm, 0.28 to 6.29, and 60 to 245 × 10^−4^ J, respectively.

Commercial wheat flours showed significantly higher levels of *P* compared to monovarietal wheat flours (*p* < 0.05). Tenacity (*P*) measures the resistance to deformation of the dough, mainly due to the presence of glutenins, while extensibility (*L*) evaluates the viscosity of the dough, conferred mainly by gliadins [38]. In the present study, commercial wheat flours were more tenacious and had significantly higher levels of *P* and glutenins compared to monovarietal wheat flours (*p* < 0.05).

A *P/L* ratio close to 1.0 is adequate for making bread, while below 0.2 is used for making short-textured products. In Indian bread wheat varieties, the *P/L* ranged from 0.66 (NI5439) to 4.91 (PBW590). In general, wheat flours with *P/L* values greater than 2 are poorly extensible and not suitable for making bread [33]. In our study, a *P/L* ratio close to 1 was only observed in commercial wheat flours. However, no statistically significant differences were observed in terms of *L* and *P/L* levels between both varieties of wheat flour.

Commercial wheat flours presented a significant increase in *W* compared to monovarietal wheat flours (*p* < 0.01). Bakery strength (*W*) is one of the best parameters to differentiate wheat flours by gluten strength and extensibility, and is associated with the energy required to inflate the dough bubble to its breaking point. According to the correlation between *W* and gluten strength, wheat flours can be classified into five classes, *W*: <100 × 10^−4^ J weak, *W*: 101–150 × 10^−4^ J moderately weak, *W*: 151–200 × 10^−4^ J moderately strong, and *W*: 201–250 × 10^−4^ J strong gluten [33]. In the present study, according to this classification, monovarietal and commercial wheat flours were moderately weak and moderately strong, respectively. Regarding *WAC* levels, no statistically significant differences were found between commercial and monovarietal wheat flours.

In commercial and monovarietal wheat flours, dry gluten was significantly correlated with the levels of protein, wet gluten, gliadins, glutenins, *L*, *P/L*, and *W* (r = −0.943, r = 0.993, r = −0.652, r = 0.652, r = 0.727, r = −0.733 and r = 0.691, *p* < 0.05, respectively). Likewise, the gluten index showed a significant correlation with the levels of moisture, falling number, *P*, and *W* (r = −0.648, r = 0.780, r = 0.691 and r = 0.652, *p* < 0.05).

Finally, multifactorial predictive models were developed to estimate the value of the analytical parameters of commercial and monovarietal wheat flour based on their rheological characteristics. Three predictive models were developed for dry gluten, gluten index, and falling number.

The regression model for the dry gluten variable explained 89.6% of the variation in this dependent variable (R^2^ adjusted = 0.896), which included two independent variables: *W* and *P*. The correlation coefficients of these variables were statistically significant (*p* < 0.001). When evaluating the semi-partial correlation coefficients, it was observed that the variables *W* (*r*_sp_ = 0.942) and *P* (*r*_sp_ = −0.904) had a high-level effect on the dry gluten variable.
(3)$$Dry\;gluten=7.020+0.039\left(W\right)-0.056\left(P\right)$$

The regression model for the dependent variable gluten index contained two independent variables: *W* and *P/L*. The correlation coefficients for all these variables were statistically significant (*p* < 0.01). The adjusted coefficient of determination (adjusted R^2^ = 0.805) indicated that the model explained 80.5% of the variation in the variable falling number. According to the values of the semi-partial correlation coefficients, the variables *W* (*r*_sp_ = 0.885) and *P/L* (*r*_sp_ = 0.831) had a high-level effect on the gluten index variable.
(4)$$Gluten\;index=24.580+0.275\left(W\right)+7.304\;\left(\frac{P}{L}\right)$$

The linear regression model for the variable falling number explained 76.1% of the variance in this dependent variable (adjusted R^2^ = 0.761). This model included the independent variables: *W*, ash, protein, gluten index, and albumins/globulins, whose regression coefficients were statistically significant (*p* < 0.05). The semi-partial correlation coefficients indicated that the variables *W* (*r*_sp_ = 0.883) and *WAC* (*r*_sp_ = −0.584) had a high-level and medium-level effect, respectively.
(5)$$Falling\;number=363.491+1.776\left(W\right)-5.489\left(WAC\right)$$

The present study has some limitations that need to be addressed. First, monovarietal wheat flours were obtained by milling on a laboratory scale. Secondly, the monovarietal wheat flours evaluated in this research were obtained from a single province of the Andean region of Peru. Third, commercial wheat that are sold in Peru are made from a mixture of imported wheat. Among the strengths of this article, this is the first study that compares the quality of commercial and monovarietal wheat flours through their physicochemical and rheological characteristics, gluten content, and wheat protein fractions. Due to the environmental crisis caused by climate change and the war between Russia and Ukraine, the demand and cost of wheat have increased significantly worldwide. Therefore, the importance of the study lies in promoting the search for new monovarietal wheat germplasms from the Andean region of Peru that are tolerant to water stress and high temperatures and make efficient use of water. Furthermore, Peruvian wheat could have other uses. In the Andean region of Peru, its internal consumption is widespread through the form of “morón”, which is a coarsely ground wheat, which has been previously peeled and toasted, widely used for the preparation of soups, which stands out for its high levels of minerals, proteins, and carbohydrates.

## 4. Conclusions

Commercial wheat flours presented significantly higher levels of physicochemical parameters such as proteins, gluten index, falling number, and glutenins compared to monovarietal wheat flours. Likewise, commercial wheat flours showed a significantly higher content of rheological parameters such as *P* and *W* compared to monovarietal wheat flours. In all the predictive models used to evaluate the quality parameters of wheat flours (dry gluten, gluten index, and falling number) based on their rheological characteristics, the baking strength (*W*) parameter was the most important. The monovarietal wheat from Peru did not present a quality for the elaboration of flours comparable to commercial wheat flours; however, through other uses, it is an important component of the diet of the Andean region of Peru.

## Figures and Tables

**Figure 1 foods-12-01789-f001:**
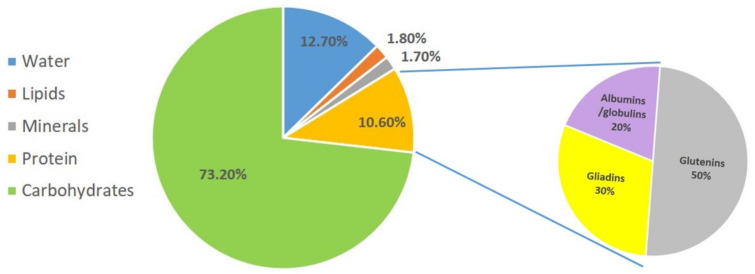
Distribution of grain components and proteins of common wheat.

**Figure 2 foods-12-01789-f002:**
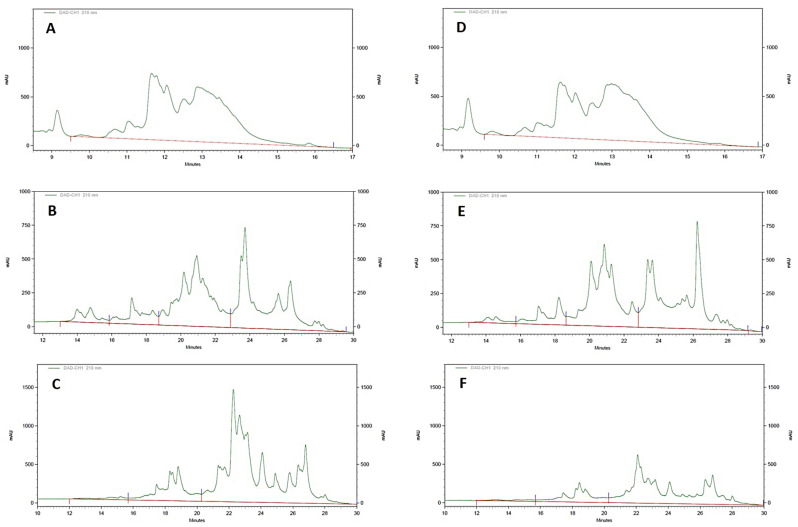
RP-HPLC chromatograms in representative samples of commercial and monovarietal wheat flour. (**A**) Albumins/globulins—Commercial flour; (**B**) Gliadins—Commercial flour; (**C**) Glutenins—Commercial flour; (**D**) Albumins/globulins—Monovarietal flour; (**E**) Gliadins—Monovarietal flour (**F**) Glutenins—Monovarietal flour.

**Table 1 foods-12-01789-t001:** Commercial and monovarietal wheat flours from Peru.

Code	Variety	Origin
WF100	Commercial	“Santa Anita” Producers Market, Lima, Peru
WF101	Commercial	“Santa Anita” Producers Market, Lima, Peru
WF102	Commercial	“Santa Anita” Producers Market, Lima, Peru
WF103	Commercial	“Santa Anita” Producers Market, Lima, Peru
WF104	Commercial	“Santa Anita” Producers Market, Lima, Peru
WF105	Commercial	“Santa Anita” Producers Market, Lima, Peru
WF106	Commercial	“Santa Anita” Producers Market, Lima, Peru
WF107	Commercial	“Santa Anita” Producers Market, Lima, Peru
Barba negra	Monovarietal	Acobamba Province, Huancavelica Department, Peru
Espelta	Monovarietal	Acobamba Province, Huancavelica Department, Peru
Candeal	Monovarietal	Acobamba Province, Huancavelica Department, Peru
Duro	Monovarietal	Acobamba Province, Huancavelica Department, Peru

WF—wheat flour.

**Table 2 foods-12-01789-t002:** Proximate analysis of commercial and monovarietal wheat flours from Peru.

Parameter	Commercial (*n* = 8)	Monovarietal (*n* = 4)	*p*-Value
Moisture (%)	13.67 ± 0.56	15.52 ± 0.49	<0.001
Ash (%)	0.60 ± 0.07	0.66 ± 0.04	0.035
Fat (%)	1.19 ± 0.30	0.29 ± 0.06	<0.001
Protein (%)	9.64 ± 0.60	8.02 ± 2.45	0.018
Carbohydrates (%)	74.91 ± 0.99	75.52 ± 2.54	0.400
Crude fiber (%)	0.12 ± 0.32	0.15 ± 0.39	0.036
Total energy (Kcal/100 g)	348.87 ± 1.51	336.76 ± 1.92	<0.001

Data are expressed as the media ± SD. Differences between groups were evaluated using a two-sample *t*-test analysis, *p* < 0.05.

**Table 3 foods-12-01789-t003:** Chromatic properties of commercial and monovarietal wheat flours from Peru.

Parameter	Commercial (*n* = 8)	Monovarietal (*n* = 4)	*p*-Value
*L**	85.11 ± 1.27	86.92 ± 1.43	<0.001
*a**	−2.45 ± 0.29	−2.48 ± 0.16	0.580
*b**	11.90 ± 1.38	10.43 ± 1.40	<0.001
C**ab*	12.15 ± 1.40	10.73 ± 1.37	<0.001
h*ab*	78.32 ± 0.90	76.46 ± 1.68	<0.001

Data are expressed as the media ± SD. Differences between groups were evaluated using a two-sample *t*-test analysis, *p* < 0.05.

**Table 4 foods-12-01789-t004:** Gluten content and falling number of commercial and monovarietal wheat flours from Peru.

Parameter	Commercial (*n* = 8)	Monovarietal (*n* = 4)	*p*-Value
Wet gluten (%)	25.89 ± 3.01	20.64 ± 10.50	0.072
Dry gluten (%)	8.78 ± 1.00	6.96 ± 3.35	0.053
Gluten index	95.94 ± 4.45	74.75 ± 23.69	0.002
Falling number (s)	418.13 ± 62.35	284.00 ± 103.14	<0.001
Albumins/globulins (%)	21.57 ± 2.47	23.96 ± 2.28	0.032
Gliadins (%)	32.70 ± 3.66	37.55 ± 3.95	0.007
Glutenins (%)	45.73 ± 5.66	38.49 ± 4.87	0.005

Data are expressed as the media ± SD. Differences between groups were evaluated using a two-sample *t*-test analysis, *p* < 0.05.

**Table 5 foods-12-01789-t005:** Rheological analysis of commercial and monovarietal wheat flours from Peru.

Parameter	Commercial (*n* = 8)	Monovarietal (*n* = 4)	*p*-Value
*WAC* (%)	55.80 ± 0.71	51.15 ± 9.96	0.380
*P* (mm)	82.00 ± 1.41	81.00 ± 36.06	0.029
*L* (mm)	62.50 ± 2.83	65.75 ± 54.20	0.233
*P/L*	1.33 ± 0.03	2.80 ± 2.58	0.431
*W* (10^−4^ J)	183.00 ± 2.83	115.00 ± 47.39	<0.001

Data are expressed as the media ± SD. Differences between groups were evaluated using a two-sample *t*-test analysis, *p* < 0.05.

## Data Availability

The data presented in this study are available on request from the corresponding author.

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
