# Peer review of "Physicochemical and Rheological Characteristics of Commercial and Monovarietal Wheat Flours from Peru"

_foods, 2023, doi:10.3390/foods12091789_

Round 1
Reviewer 1 Report
Dear Authors,
The objective of this study was to evaluate the baking quality of commercial and monovarietal wheat flours used mostly in the Peruvian baking industry by evaluating their physicochemical and rheological parameters. In terms of testing the baking quality of flour, the tests cited in the publication are indirect methods. The main criterion for assessing the baking quality of flour is the baking test. During baking, we receive information about the behavior of the dough during preparation and further processing, as well as the quality of the bread obtained. I mean its volume, organoleptic characteristics, textural properties. During pre-baking, we can evaluate the specific characteristics of the flour, which we are unable to define using indirect methods. The presented results do not refer to the part of the topic of the work - "...baking quality..." Please complete the work with the results of the baking test and analyzes of baked bread. The work will be complete and valuable. Very well started research and article, but reading the article you can feel that the next part is missing.
Author Response
Point 1: The objective of this study was to evaluate the baking quality of commercial and monovarietal wheat flours used mostly in the Peruvian baking industry by evaluating their physicochemical and rheological parameters. In terms of testing the baking quality of flour, the tests cited in the publication are indirect methods. The main criterion for assessing the baking quality of flour is the baking test. During baking, we receive information about the behavior of the dough during preparation and further processing, as well as the quality of the bread obtained. I mean its volume, organoleptic characteristics, textural properties. During pre-baking, we can evaluate the specific characteristics of the flour, which we are unable to define using indirect methods. The presented results do not refer to the part of the topic of the work - "...baking quality..." Please complete the work with the results of the baking test and analyzes of baked bread. The work will be complete and valuable. Very well started research and article, but reading the article you can feel that the next part is missing.
Response 1: The title and objective of the study were modified according to the study needs, to evaluate the physicochemical and rheological characteristics of commercial and monovarietal wheat flours from Peru.
Reviewer 2 Report
Comment#1: The title of the study is confusing. The objective of this study was to evaluate the baking quality of commercial and monovarietal wheat flour. Physicochemical and rheological tests were used to evaluate the wheat quality. The title shows that Physicochemical and rheological tests are associated with wheat properties that are already known. Please change the title according to the study's needs.
Comment#2: The main introduction does not fully explain the need for the evaluation of protein content in wheat in Peru. It has been mentioned just in one line. Please provide more information on the need for it with some good references.
Comment#3: Gliadin is an integral component of Gluten. If gluten was significantly higher in commercial wheat. how come gliadin was higher in monovarietal wheat? Lines 28-30
Comment#4: What is the current compound annual rate? Line no. 40
Comment#5: Please elaborate on the paragraph. e.g., collection of commercial and monovarietal flour samples. Section 2.1
Comment#6: Why were 8 wheat varieties selected? Are they frequently used or there are only these 8 varieties available in Peru? Line no 88-89
Comment#7: Section 2.2 could be the ‘measurement wheat of properties’. Successive sections would be section 2.2.1 explaining the respective tests.
Comment#8: Please summarize the AACC method to determine moisture, ash, fat, and protein content, gluten analysis method, and α-amylase activity method briefly.
Comment#9: In the current style, it is not easy to understand the actual results of the study. Therefore, please write the results and discussion in separate sections. The MDPI also suggests the same style.
Comment#10: In the discussion section please also mention the limitations and strengths of the study.
Minor comments
Comment#1: What is the word ‘input’ stands for? Line no 20
Comment#2: What is the need to check the quality of wheat? Line no. 21
Comment#3: The word ‘can be’ makes the line weak. These tests can be used, or they are used to determine the wheat quality. Line no. 21
Comment#4: Position (latitude and longitude) is not needed. Line no. 90
Comment#5: Please define, what is monovarietal wheat. Line no. 92
Comment#6: What are (1) and (2) in the given formulas? Lines no. 117-120
Comment#7: Please use the symbol for Alpha (α-). Line no 127
Comment#8: Mention the full form of SPSS software.
Comment#9: Line no. 173 should go at the last of the paragraph. i.e., line no 177
Suggestion
A schematic diagram could be a good idea to show wheat protein percentage. Lines no. 65-71
Author Response
Point 1: The title of the study is confusing. The objective of this study was to evaluate the baking quality of commercial and monovarietal wheat flour. Physicochemical and rheological tests were used to evaluate the wheat quality. The title shows that Physicochemical and rheological tests are associated with wheat properties that are already known. Please change the title according to the study's needs.
Response 1: The title and objective were modified according to the needs of the study.
Point 2: The main introduction does not fully explain the need for the evaluation of protein content in wheat in Peru. It has been mentioned just in one line. Please provide more information on the need for it with some good references.
Response 2: In line 74, the importance of evaluating the protein content in wheat flours was added.
Point 3: Gliadin is an integral component of Gluten. If gluten was significantly higher in commercial wheat. how come gliadin was higher in monovarietal wheat? Lines 28-30.
Response 3: Commercial wheat flours presented higher levels of wet gluten and dry gluten compared to monovarietal wheat flours; however, no significant differences were found between various varieties of flours. Considering the total gluten content of commercial and monovarietal wheat flours evaluated by RP-HPLC, gliadins (expressed as a percentage) were significantly higher in monovarietal wheat flours compared to commercial wheat flours.
Point 4: What is the current compound annual rate? Line no. 40
Response 4: The required data was added.
Point 5: Please elaborate on the paragraph. e.g., collection of commercial and monovarietal flour samples. Section 2.1
Response 5: Change was made as suggested.
Point 6: Why were 8 wheat varieties selected? Are they frequently used or there are only these 8 varieties available in Peru? Line no 88-89.
Response 6: The most economically important commercial and monovarietal wheat flours in Peru were selected. Changes were made as suggested.
Point 7: Section 2.2 could be the ‘measurement wheat of properties’. Successive sections would be section 2.2.1 explaining the respective tests.
Response 7: Changes were made as suggested.
Point 8: Please summarize the AACC method to determine moisture, ash, fat, and protein content, gluten analysis method, and α-amylase activity method briefly.
Response 8: Regarding the measurement of moisture, ash, fat, and protein, the AOAC 925.10, 923.03, 922.06 and 920.87 methods were used, respectively. The requested methods are described below:
AOAC Official Method 925.10. Solids (Total) and Moisture in Flour. Air Oven Method. In cooled and weighed dish (provided with cover), 925.09A(a) (see 32.1.02), previously heated to 130 ± 3 °C, accurately weigh ca 2 g well-mixed sample. Uncover sample, and dry dish, cover, and contents 1 h in oven provided with opening for ventilation and maintained at 130 ± 3 °C (1 h drying period begins when oven temperature is actually 130 °C). Cover dish while still in oven, transfer to desiccator, and weigh soon after reaching room temperature. Report flour residue as total solids and loss in weight as moisture (indirect method).
AOAC Official Method 923.03 Ash of Flour. Direct Method. Weigh 3–5 g well-mixed sample into shallow, relatively broad ashing dish that has been ignited, cooled in desiccator, and weighed soon after reaching room temperature. Ignite in furnace at ca 550 °C (dull red) until light gray ash results, or to constant weight. Cool in desiccator and weigh soon after reaching room temperature. Reignited CaO is satisfactory drying agent for desiccator.
AOAC Official Method 922.06 Fat in Flour. Acid Hydrolysis Method. Place 2 g sample in 50 mL beaker, add 2 mL alcohol, and stir to moisten all particles to prevent lumping on addition of acid. Add 10 mL HCl (25 + 11), mix well, set beaker in H2O bath held at 70–80 °C, and stir at frequent intervals during 30–40 min. Add 10 mL alcohol and cool. Transfer mixture to Mojonnier fat-extraction apparatus. Rinse beaker into extraction tube with 25 mL ether, added in 3 portions; stopper flask (with glass, cork, Neoprene, or other synthetic rubber stopper not affected by solvents) and shake vigorously 1 min. Add 25 mL redistilled petroleum ether (bp <60°) and again shake vigorously 1 min. Let stand until upper liquid is practically clear, or centrifuge 20 min at ca 600 rpm. Draw off as much as possible of ether-fat solution through filter consisting of cotton pledged packed just firmly enough in funnel stem
to let ether, pass freely into weighed 125 mL beaker-flask containing porcelain chips or broken glass. Before weighing beaker-flask, dry it and similar flask as counterpoise in oven at 100 °C; then let stand in air to constant weight. Re-extract liquid remaining in tube twice, each time with only 15 mL of each ether. Shake well on addition of each ether. Draw off clear ether solutions through filter into same flask as before and wash tip of spigot, funnel, and end of funnel stem with few mL of mixture of the 2 ethers in equal volumes, free from suspended H2O. Evaporate ethers slowly on steam bath; then dry fat in oven at 100 °C to constant weight (ca 90 min). Remove flask and counterpoise from oven, let stand in air to constant weight (ca 30 min), and weigh. (Owing to size of flask and nature of material, there is less error by cooling in air than by cooling in desiccator.) Correct this weight by blank determination on reagents used. Report as % fat by acid hydrolysis.
AOAC Official Method 920.87 Protein (Total) in Flour. Place weighed sample (0.7–2.2 g) in digestion flask. Add 0.7 g HgO or 0.65 g metallic Hg, 15 g powdered K2SO4 or anhydrous Na2SO4, and 25 mL H2SO4. If sample >2.2 g is used, increase H2SO4 by 10 mL for each g sample. Place flask in inclined position and heat gently until frothing ceases (if necessary, add small amount of paraffin to reduce frothing); boil briskly until solution clears and then 30 min longer (2 h for samples containing organic material). Cool, add ca 200 mL H2O, cool <25°, add 25 mL of the sulfide or thiosulfate solution, and mix to precipitate Hg. Add few Zn granules to prevent bumping, tilt flask, and add layer of NaOH without agitation (For each 10 mL H2SO4 used, or its equivalent in diluted H2SO4, add 15 g solid NaOH or enough solution to make contents strongly alkaline). (Thiosulfate or sulfide solution may be mixed with the NaOH solution before addition to flask.) Immediately connect flask to distilling bulb on condenser, and, with tip of condenser immersed in standard acid and 5–7 drops indicator in receiver, rotate flask to mix contents thoroughly; then heat until all NH3 had distilled (150 mL distillate). Remove receiver, wash tip of condenser, and titrate excess standard acid in distillate with standard NaOH solution. Correct for blank determination on reagents.
% N = [(mL standard acid normality acid) – (mL standard NaOH normality NaOH)] x 1.4007/g sample
Multiply % N by 5.7 to obtain % protein.
AACC 38-12.02 Wet Gluten, Dry Gluten, and Gluten Index
This method is based on washing a sample of flour or whole meal flour by a Glutomatic gluten washer to obtain gluten, and centrifugation in a gluten centrifuge by means of a specially built sieve under controlled standardized conditions. This sieve allows the collection of both the gluten that passes through and that which remains on the sieve. The total weight of gluten is defined as the amount of wet gluten. The percentage of wet gluten that remains without passing through the sieve after centrifugation is defined as the gluten index. The amount of gluten dried between two teflon-coated plates is defined as dry gluten.
AACC 56-81B Flour Falling Numbers
The Falling Number method is based on the principle of the rapid gelatinization of an aqueous suspension of flour in boiling water and the subsequent measurement of the liquefaction of the starch by the action of α-amylase in the sample.
Point 9: In the current style, it is not easy to understand the actual results of the study. Therefore, please write the results and discussion in separate sections. The MDPI also suggests the same style.
Response 9: The Results and Discussion section was modified to allow a better understanding of the study, a combined Results and Discussion section is a valid format accepted by Foods.
Point 10: In the discussion section please also mention the limitations and strengths of the study.
Response 10: In line 398, a paragraph on the limitations and strengths and of the study was added.
Minor comments
Point 1: What is the word ‘input’ stands for? Line no 20.
Response 1: Change was made as suggested.
Point 2: What is the need to check the quality of wheat? Line no. 21.
Response 2: Change was made as suggested.
Point 3: The word ‘can be’ makes the line weak. These tests can be used, or they are used to determine the wheat quality. Line no. 21.
Response 3: Change was made as suggested.
Point 4: Position (latitude and longitude) is not needed. Line no. 90.
Response 4: Position (latitude and longitude) was removed.
Point 5: Please define, what is monovarietal wheat. Line no. 92.
Response 5: The definition of monovarietal wheat was added.
Point 6: What are (1) and (2) in the given formulas? Lines no. 117-120.
Response 6: In line 137, it is described that C*ab and hab correspond to chroma and hue angle, respectively.
Point 7: Please use the symbol for Alpha (α-). Line no 127.
Response 7: Change was made as suggested.
Point 8: Mention the full form of SPSS software.
Response 8: Change was made as suggested.
Point 9: Line no. 173 should go at the last of the paragraph. i.e., line no 177.
Response 9: Change was made as suggested.
Point 10: Suggestion. A schematic diagram could be a good idea to show wheat protein percentage. Lines no. 65-71.
Response 10: Figure 1 was added.
Reviewer 3 Report
After a careful review of the manuscript entitled “Physicochemical and rheological characteristics associated with 2 the baking quality of commercial and monovarietal wheat 3 flours from Peru” here are few minor recommendations. Overall, it is a well written and informative article, but I suggest some revisions in the article. Please address the following points in the revised manuscript.
1. Please add one are more sentences about the novelty in the abstract.
2. Objectives of the study should be clearly mentioned in the manuscript with more details.
3. In the abstract, also mention the major findings and best one results.
4. In the material and methods; change the heading of physic-chemical analysis with proximate analysis.
5. What is the extraction rate of flour?
6. The storage temperature of the flour in -20. Justify your sentence regarding too low storage temperature for flour. And also add any reference.
7. In the introduction, a lot of grammar and sentences mistakes are observed, try to remove these.
8. Revise the format of the Tables given, Follow the author guide lines for figures and tables in the manuscript.
9. The figures should be changed as these are difficult to understand and use high pixels images.
10. The result needs more details discussion regarding the trend and complete mechanism.
11. At the last, conclusion should be short and relevant to the study.
12. The references should be updated and relevant to the study
Author Response
Point 1: Please add one are more sentences about the novelty in the abstract.
Response 1: Changes were made as suggested.
Point 2: Objectives of the study should be clearly mentioned in the manuscript with more details.
Response 2: The objectives were modified according to the needs of the study.
Point 3: In the abstract, also mention the major findings and best one results.
Response 3: The abstract was modified as suggested.
Point 4: In the material and methods; change the heading of physic-chemical analysis with proximate analysis.
Response 4: Change was made as suggested.
Point 5: What is the extraction rate of flour?
Response 5: In commercial wheat flours, the flour extraction rate is approximately 78-80%. In monovarietal wheat flours made with laboratory scale milling, the flour extraction rate decreases to approximately 70 to 75%.
Point 6: The storage temperature of the flour in -20. Justify your sentence regarding too low storage temperature for flour. And also add any reference.
Response 6: Wheat flours have a shelf life of approximately 6 months. Because these were collected progressively, the wheat flours were kept at -20 °C in vacuum-sealed bags (not permeable to oxygen and moisture) until the analyzes were performed.
Point 7: In the introduction, a lot of grammar and sentences mistakes are observed, try to remove these.
Response 7: Changes were made as suggested.
Point 8: Revise the format of the Tables given, Follow the author guide lines for figures and tables in the manuscript.
Response 8: The format of the tables and figures was revised according to the instructions for authors.
Point 9: The figures should be changed as these are difficult to understand and use high pixels images.
Response 9: The resolution of the figures was updated.
Point 10: The result needs more details discussion regarding the trend and complete mechanism.
Response 10: The discussion was modified as suggested.
Point 11: At the last, conclusion should be short and relevant to the study.
Response 11: The conclusion was modified as suggested.
Point 12: The references should be updated and relevant to the study.
Response 12: More updated and relevant references to the study were added.
Reviewer 4 Report
This study evaluates the bread quality of wheat flours of 8 commercial and 4 monovarietal wheat flours from Peru using a variety of physicochemical and analytical methods. Reporting the wheat flour characteristics of Peruvian wheat is important in valorizing underutilized wheat varieties. However, in the current form the manuscript demonstrates several deficiencies specifically in the introduction, methodology and results. Some comments are included:
Abstract and introduction:
1. Please provide an explanation of the advantage and novelty of using wheat flours from Peru. Although the wheat quality is regarded to be poor, are the varieties examined more sustainable, drought tolerant, etc..?
2. Either in the abstract or in the introduction, please clarify what is meant by poor production (yield, wheat quality, or processing).
3. In the introduction (lines 82 to 85), the aims of the paper or hypothesis is not clearly provided.
Materials and Methods: In this current form, the methods described are not sufficient for these experiments to be duplicated. The major issues are as follows:
1. Page 2, Line 88 and Table 1: Of the 8 commercial wheat flours, were these blended flours and/or bread flour or durum wheat? The HTES### coding is not clear.
2. Page 2, Lines 88 to 92: The study design is not clear. Milling plays a critical role in wheat flour functionality. In fact, the authors mention this in line 194. Although the Chopin CD1 mill is often used for lab scale milling of wheat, what controls were used to examine the effect of the different mills used across the different commercial sources of flour and the flour produced by the lab scale mill?
3. Page 2, Line 91: In the preparation of the flours for milling, did they undergo any conditioning? If so, please provide those details.
4. Page 2, Line 92: Please provide additional details of the flour used for the experiments such as the particle size of the commercial flours and if the flours were whole meal?
5. Page 2, line 93: Please elaborate when analysis was performed (before or after storage at -20°C) and their duration of storage.
6. Page 3, Lines 99: Please provide a list of materials for the physicochemical analysis and gluten analysis.
7. Page 3, Lines 99 to 103 and Table 2: What were the number of replicates for these samples or were they single analysis for each flour?
8. Page 4, Lines 161: Please add details of the number of replicates that were conducted for each test to provide clarity of the values that are being reported (mean ± SD).
9. Please clarify the groups that were compared, and the post hoc test used for the ANOVA analysis.
Results and Discussion:
1. Page 5, Lines 199 to 200: This sentence “Commercial wheat flours presented significantly higher protein levels compared to commercial wheat…”
2. Page 5 Table 2: The footnote indicates that a one-way ANOVA was performed but there are only two columns (commercial versus monovarietal) is reported. The current presentation of table 2 is not clear and is misleading. There is quite a bit of overlap in the reported mean and standard deviation in those endpoints that show significant p-values. Please provide specific details of the significant differences found and their respective mean separation.
3. Page 5, Lines 196 to 199: Please expand further on the impact of higher crude fiber on baking quality.
4. Page 8, Figure 1: Reporting side-by side examples of a representative commercial and a representative monovarietal chromatograms of albumins/globulins, glutenins, and gliadins would elevate this figure.
5. Tables 2 through Table 5: Although the footnotes in the tables indicate that a one-way ANOVA analysis was conducted between groups, only two groups are shown please reformat tables to reflect the one-way ANOVA and specify in the footnotes the post hoc analysis used.
The conclusion summarized findings well, but the strength of this paper
should not only be focused on the superior features of the commercial flour
but perhaps also consider/mention the bread quality attributes of the
monovarietal wheat such as their falling number and crude dietary fiber.
Increasing dietary fiber intake continues to grow in popularity.
Author Response
Abstract and introduction
Point 1: Please provide an explanation of the advantage and novelty of using wheat flours from Peru. Although the wheat quality is regarded to be poor, are the varieties examined more sustainable, drought tolerant, etc..?
Response 1: In line 402, the following paragraph was added: “Among the strengths of this article, it can be mentioned that it is the first study that compares the quality of commercial and monovarietal wheat flours through their physicochemical and rheological characteristics, gluten content and wheat protein fractions. Due to the environmental crisis caused by climate change, and the war between Russia and Ukraine, the demand and cost of wheat have increased significantly worldwide. Therefore, the importance of the study lies in promoting the search for new monovarietal wheat germplasms from the Andean region of Peru that are tolerant to water stress, high temperatures and make efficient use of water. Likewise, Peruvian wheat could have other uses. In the Andean region of Peru, its internal consumption is widespread through the form of "morón", which is a coarsely ground wheat, which has been previously peeled and toasted, widely used for the preparation of soups, which stands out for its high levels of minerals, proteins and carbohydrates”.
Point 2: Either in the abstract or in the introduction, please clarify what is meant by poor production (yield, wheat quality, or processing).
Response 2: Changes were made as suggested.
Point 3: In the introduction (lines 82 to 85), the aims of the paper or hypothesis is not clearly provided.
Response 3: The aim of the study was modified as suggested.
Materials and Methods
Point 4: Page 2, Line 88 and Table 1: Of the 8 commercial wheat flours, were these blended flours and/or bread flour or durum wheat? The HTES### coding is not clear.
Response 4: Peru is a wheat importing country, the commercial wheat flours that are sold in Peru are a mixture of wheat coming mainly from Canada, Argentina or the USA. Each mill in Peru manages its own mixtures to obtain wheat flour.
HTES was changed to WF (wheat flour).
Point 5: Page 2, Lines 88 to 92: The study design is not clear. Milling plays a critical role in wheat flour functionality. In fact, the authors mention this in line 194. Although the Chopin CD1 mill is often used for lab scale milling of wheat, what controls were used to examine the effect of the different mills used across the different commercial sources of flour and the flour produced by the lab scale mill?
Response 5: Although the conditions for obtaining commercial and monovarietal wheat flours were different, we worked with commercial and monovarietal wheat flours that had approximately a similar particle size. These flours were characterized by physicochemical and rheological analysis, as well as the gluten content and wheat protein fractions.
Point 6: Page 2, Line 91: In the preparation of the flours for milling, did they undergo any conditioning? If so, please provide those details.
Response 6: Previously, the wheat is cleaned to separate stems, dust, stones and damaged grains. Subsequently, the conditioning of the grain is carried out at 16% water.
Point 7: Page 2, Line 92: Please provide additional details of the flour used for the experiments such as the particle size of the commercial flours and if the flours were whole meal?
Response 7: In the present study, whole wheat flours were not used. The commercial and monovarietal wheat flours presented a similar particle size, the percentage retained in a 180 um mesh for commercial and monovarietal flours was 0.841% and 0.173%, respectively.
Point 8: Page 2, line 93: Please elaborate when analysis was performed (before or after storage at -20°C) and their duration of storage.
Response 8: In line 113, it is indicated that the analyzes were performed after storage at -20 °C.
Point 9: Page 3, Lines 99: Please provide a list of materials for the physicochemical analysis and gluten analysis.
Response 9: The list of materials for the physicochemical analysis and gluten analysis was added.
Point 10: Page 3, Lines 99 to 103 and Table 2: What were the number of replicates for these samples or were they single analysis for each flour?
Response 10: The required information was added.
Point 11: Page 4, Lines 161: Please add details of the number of replicates that were conducted for each test to provide clarity of the values that are being reported (mean ± SD).
Response 11: Data represent the mean ± standard deviation of duplicate samples. Statistical procedures based on Bayesian theory indicate that one of the conditions is that chemical or physical analyses should be carried out in triplicate, or at least duplicate (Aparicio R, García-González DL. 2013. Olive oil characterization and traceability. In Aparicio R, Harwood J eds. Handbook of Olive Oil: Analysis and Properties, pp. 431-472. Springer Science+Business Media, New York).
Point 12: Please clarify the groups that were compared, and the post hoc test used for the ANOVA analysis.
Response 12: Physicochemical and rheological characteristics were compared between commercial and monovarietal wheat flours using a two-sample t-test analysis, p < 0.05.
Results and Discussion:
Point 13: Page 5, Lines 199 to 200: This sentence “Commercial wheat flours presented significantly higher protein levels compared to commercial wheat…”
Response 13: This sentence was modified: “Commercial wheat flours presented significantly higher protein levels compared to movarietals wheat flours (p < 0.05)”.
Point 14: Page 5 Table 2: The footnote indicates that a one-way ANOVA was performed but there are only two columns (commercial versus monovarietal) is reported. The current presentation of table 2 is not clear and is misleading. There is quite a bit of overlap in the reported mean and standard deviation in those endpoints that show significant p-values. Please provide specific details of the significant differences found and their respective mean separation.
Response 14: The differences between commercial and monovarietal wheat flours were evaluated by means of a two-sample t-test analysis, at a significance level of 0.05.
Point 15: Page 5, Lines 196 to 199: Please expand further on the impact of higher crude fiber on baking quality.
Response 15: In line 224, this information was added.
Point 16: Page 8, Figure 1: Reporting side-by side examples of a representative commercial and a representative monovarietal chromatograms of albumins/globulins, glutenins, and gliadins would elevate this figure.
Response 16: The figure was modified as suggested.
Point 17: Tables 2 through Table 5: Although the footnotes in the tables indicate that a one-way ANOVA analysis was conducted between groups, only two groups are shown please reformat tables to reflect the one-way ANOVA and specify in the footnotes the post hoc analysis used.
Response 17: We agreed with the reviewer. As indicated in tables 2 to 5, to compare the means of both groups, a two-sided t-test analysis was performed.
Round 2
Reviewer 4 Report
I commend the authors for the revisions. The added discussion of the limitations of this manuscript (lines 485 to 500) was an excellent addition.